# DARTS: Distribution-Aware Active Rollout Trajectory Shaping for Accelerating LLM Reinforcement Learning

Yujie Wang [*1]   Siwei Chen [*1]   Longzan Luo [*1]   Xinyi Liu [1]   Xupeng Miao [1]   Fangcheng Fu [2]   Bin Cui [13]

## Abstract

Reinforcement Learning (RL) has become pivotal for improving model capabilities yet suffers from rollout efficiency bottlenecks due to the long-tail response length distribution. While existing works mitigate the impact of long tails via prompt-level tail scheduling, we focus on the root source of inefficiency: the distribution itself. Specifically, we characterize the long-tail distribution at a finer granularity, identifying intra-prompt long tails, and revealing that they frequently consist of ineffective verbosity. To address this, we propose a novel paradigm of *active distribution shaping* to shape the rollout distribution towards conciseness and certainty, thereby fundamentally resolving tail-induced overheads. We achieve this through a *distribution-aware trajectory sampling* mechanism, which selects trajectories from a redundant exploration space for each prompt, and an *adaptive redundancy allocation* scheme to maximize both shaping effectiveness and system efficiency. Experiments demonstrate significant acceleration over state-of-the-art systems by up to $1.77\times$ without compromising model performance. Source code available at: URL.

## 1. Introduction

Reinforcement Learning (RL) has catalyzed the development of advanced LLMs (Large Language Models), such as OpenAI o-series (Jaech et al., 2024; OpenAI, 2025), Gemini 3 (Google, 2025), Claude 4.5 (Anthropic, 2025b;a), Grok 4 (xAI, 2025), DeepSeek-R1 (Guo et al., 2025),

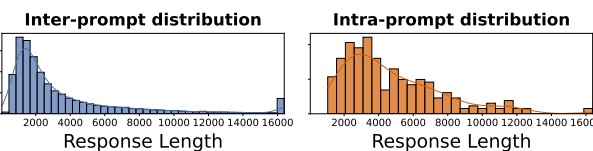

*Figure 1.* Inter-prompt and intra-prompt rollout trajectory length distribution. (Qwen3-30B-A3B on DAPO-MATH dataset)

Qwen3 (Yang et al., 2025), Kimi-K2 (Team et al., 2025a) and so on (Li et al., 2025). By shifting the focus from pre-training scaling to inference-time scaling, RL has empowered state-of-the-art models with robust reasoning capabilities, enabling them to tackle complex mathematical problems (Shao et al., 2024), coding tasks (Luo et al., 2025; Zhang et al., 2024), tool usage (Feng et al., 2026; Xu et al., 2025), database intelligent applications (Zhou et al., 2024) and intricate agentic tasks (Team et al., 2025b).

RL for LLMs (Schulman et al., 2017; Shao et al., 2024) typically comprises two major phases: *rollout* and *training*. Empirically, the rollout phase accounts for over 70% of the total RL training time, constituting the primary system bottleneck (Guan et al., 2024; Zhang et al., 2025). This stems from a severe long-tail distribution in rollout trajectory lengths (Zhong et al., 2025; Gao et al., 2026; Zhou et al., 2025), where a small fraction of prompts trigger ultra-long trajectories, often exceeding the median length by $5$–$10\times$ and short responses by over $20\times$, as shown in Fig. 1 (left). This long-tail distribution imposes severe system overheads. For one thing, these extremely long trajectories consume excessive computational resources, leading to inefficiency. For another, in synchronous on-policy RL systems (Sheng et al., 2025; Wang et al., 2025), the longest response blocks the entire batch from entering the training phase, resulting in significant device under-utilization.

Recent literature has attempted to mitigate the system bottleneck caused by long tails, primarily through *prompt-level tail scheduling*. Strategies like Tail Batching (Gao et al., 2026) speculatively over-sample prompts (sampling $N'$ prompts, where $N' > N$). Once the target $N$ prompts complete generation, the remaining ongoing prompts (typically, the long tails) are halted and deferred to a dedicated queue for batched re-computation later. Similarly, Partial

---

[*]Equal contribution  [1]School of Computer Science & Beijing Key Laboratory of Software and Hardware Cooperative Artificial Intelligence Systems, Peking University, Beijing, China [2]School of Artificial Intelligence, Shanghai Jiao Tong University, Shanghai, China [3]Institute of Computational Social Science, Peking University (Qingdao), Qingdao, China. Correspondence to: Bin Cui <cui.bin@pku.edu.cn>.

*Proceedings of the 43$^{rd}$ International Conference on Machine Learning*, Seoul, South Korea. PMLR 306, 2026. Copyright 2026 by the author(s).

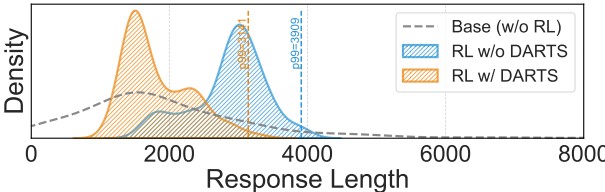

*Figure 2.* Rollout length distribution within a single prompt, w/o and w/ distribution shaping. P99 denotes the $99^{th}$ percentile.

Rollout (Team et al., 2025b; Zhou et al., 2025) speculatively over-samples and truncates the unfinished long-tail prompts when sufficient data is collected, resuming them in subsequent steps. Fundamentally, these approaches identify the *inter-prompt long-tail distribution* (Fig. 1, Left), which stems from the variance across different prompts. They rely on scheduling schemes to defer the execution of heavy-tailed prompts, aiming to prevent GPU resource underutilization.

Nevertheless, these works only try to harness the latency of long trajectories via scheduling approaches, yet fail to resolve this challenge from the root cause, namely the long-tail distribution itself. Therefore, in this work, we shift our focus to optimizing the rollout distribution.

To achieve so, we first characterize the long-tail phenomenon at a finer granularity, and find that the rollout length distribution for even a *single* prompt exhibits a pronounced long tail. We identify this as *intra-prompt long tail* distribution, which evidences that such a long-tail distribution stems from the model's inherent characteristic, rather than the prompts. Subsequently, we further inspect that these tails are not always beneficial: a substantial portion consists of verbose and ineffective trajectories, which impede training efficiency and degrade reasoning conciseness. This indicates that the model's inherent rollout length distribution is often suboptimal, wasting computational resources on low-quality trajectories.

These observations motivate us to tackle the long-tail bottleneck from a distribution-centric perspective. In particular, instead of merely scheduling around the long tails, we aim to *shape the rollout length distribution to be more concise and compact.* Fig. 2 exemplifies the effect of distribution shaping. The blue curve denotes the original distribution derived from standard RL with GRPO, and the orange curve represents our optimization target: a distribution concentrated in the concise and short-length region.

In this work, we introduce DARTS, an efficient RL framework designed to mitigate the long-tail bottleneck and accelerate the RL training pipeline via **D**istribution-aware **A**ctive **R**ollout **T**rajectory **S**haping. The core insight is *active distribution shaping*, which actively shapes the rollout distribution towards *conciseness* and *certainty*, thereby

fundamentally mitigating tail-induced overheads. Specifically, we propose a *distribution-aware trajectory sampling* mechanism. By leveraging a *dual-end length sampling* strategy, this mechanism selectively samples trajectories from a redundantly-generated exploration space for each prompt. Complementing this, we devise a *variance-based adaptive redundancy allocation* scheme, which adaptively adjusts exploration budgets to strike a balance between shaping effectiveness and system efficiency. To further maximize efficiency, we propose a series of system-level optimizations, including *variance-guided tail pruning* with *proactive early stopping* scheme, alongside a *token-level streaming* pipeline. Extensive experiments across mainstream RL algorithms and datasets demonstrate that our method achieves up to $1.77\times$ end-to-end acceleration compared to state-of-the-art systems, without compromising model performance. Our contributions are summarized as follows:

- We empirically characterize the long-tail distribution, and propose a novel paradigm of *active distribution shaping* from the distribution-centric perspective.
- We propose a *distribution-aware trajectory sampling* and *adaptive redundancy allocation* scheme, and implement a novel efficient RL framework DARTS.
- Extensive experiments demonstrate the efficiency and effectiveness of our approach and framework.

## 2. Related Work

### 2.1. Reinforcement Learning for LLMs

**The Evolution of RL for LLMs.** Reinforcement Learning (RL) has evolved from primarily aligning models with human preferences (RLHF) (Schulman et al., 2017) to driving the latest *inference-time scaling* paradigm (Jaech et al., 2024). Unlike standard next-token-prediction objective in pre-training, RL incentivizes models to explore vast solution spaces, facilitating the emergence of self-correction and long Chain-of-Thought (CoT) (Wei et al., 2022). This evolution empowers state-of-the-art reasoning models, including OpenAI o-series (Jaech et al., 2024), Gemini (Google, 2025), Claude (Anthropic, 2025b), with robust capabilities in logic-intensive domains such as mathematics and coding.

**The RL Training Pipeline.** RL algorithms for LLMs, e.g., GRPO (Shao et al., 2024), DAPO (Yu et al., 2026), operating through an iterative process of *rollout* and *training*. During the *rollout phase*, the current policy interacts with the environment to generate trajectories for a batch of prompts (e.g., mathematical problems), which are evaluated against specific verifiers to derive reward signals. In the subsequent *training phase*, the model parameters are optimized based on trajectories and their rewards. Crucially, these algorithms enforce strict on-policy semantics, requiring that the rollout parameters remain strictly synchronized with the training

*Table 1.* Summary of notations used in this paper.

| Symbol | Description |
|--------|-------------|
| $q_i$ | Input prompt sampled from dataset $\mathcal{P}(Q)$ |
| $o_i^j$ | The $j^{th}$ output response of prompt $q_i$ |
| $l_i^j$ | Response length of the rollout output $o_i^j$ |
| $M$ | Group size used for policy update |
| $M_i'$ | Redundant rollout size for prompt $q_i$ ($M_i' \geq M$) |
| $N$ | Batch size of sampled prompts for each iteration |
| $A(o_i^j)$ | Advantage estimate for the response $o_i^j$ |
| $r_i^j$ | Reward score for query $q_i$ and response $o_i^j$ pair |

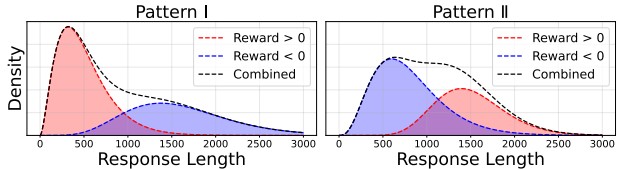

*Figure 3.* Illustration of distinct rollout distribution patterns.

from the policy model $\pi_\theta$, estimating the gradient as:

$$J(\theta) = \mathbb{E}_{q_i \sim \mathcal{P}(Q), \{o_i^j\} \sim \pi_\theta(\cdot|q_i)} \left[ \frac{1}{M} \sum_{j=1}^{M} A(o_i^j) \cdot \nabla_\theta \log \pi_\theta(o_i^j|q_i) \right]$$

(1)

Here, $\nabla_\theta \log \pi_\theta(o_i^j|q_i)$ represents the gradient of log-probability, and $A(o_i^j)$ is the advantage computed by normalizing rewards within the group:

$$A(o_i^j) = \frac{r_i^j - \mu(\mathcal{Y}_i)}{\sigma(\mathcal{Y}_i)}$$

(2)

where $r_i^j$ is the reward of output $o_i^j$ and prompt $q_i$, and $\mu(\mathcal{Y}_i)$ and $\sigma(\mathcal{Y}_i)$ denote the mean and standard deviation of the group rewards $\{r_i^j\}_{j=1}^{M}$.

### 3.2. Observations and Motivations

We analyze RL rollout characteristics, focusing on the rollout length distribution. We characterize long tails in a finer-grained granularity of *intra-prompt long tails*, and conduct an in-depth analysis of the tail patterns, motivating *distribution shaping* to mitigate these distributional inefficiencies.

#### 3.2.1. PREVALENCE OF INTRA-PROMPT LONG TAILS

Recent works (Team et al., 2025b; Zhou et al., 2025; Gao et al., 2026) primarily employ prompt scheduling approaches to tackle long-tail bottlenecks caused by prompt diversity. However, these methods typically mitigate the latency of long tails, largely overlooking the root cause, the underlying rollout distribution. In this work, we shift our focus to optimizing the rollout distribution itself, characterizing long tails at a finer granularity. Specifically, we observe significant length variance and skewed distribution *within* the rollouts of a single prompt (Fig. 1, Right): the maximum length can exceed the mean by over $10\times$. We term this the *intra-prompt long-tail distribution*, which reflects the model's inherent rollout distribution for each prompt. Orthogonal to scheduling, we aim to directly mitigate the long tails from this fine-grained intra-prompt distribution perspective, thereby eliminating excessive computational overheads and enhancing rollout efficiency.

#### 3.2.2. DECONSTRUCTING THE TAIL PATTERNS

A critical question arises: *Are all of these expensive long tails necessary?* To assess their value, we analyze the length-

---

version. This two-phase cycle repeats until convergence, with the rollout phase constituting the primary computational bottleneck due to long-tail phenomenon (Section 1).

### 2.2. Rollout Long-tail Optimization Techniques

**Prompt-level Tail Scheduling.** Recent literature mitigates the long-tail bottleneck primarily through *prompt-level tail scheduling*. RollPacker (Gao et al., 2026) introduces Tail Batching, which speculatively over-samples prompts (sampling $N' > N$) to identify long-tail outliers. Once $N$ prompts complete rollout, instead of blocking the pipeline, it defers the unfinished prompts to dedicated queues for batched re-computation later. This ensures that the majority of training steps remain load-balanced while preserving strict on-policy semantics. Similarly, Partial Rollout (Zhou et al., 2025; Team et al., 2025b) employs over-sampling but adopts a truncate-and-resume strategy. Once the target sample count is collected, it terminates the batch and buffers the unfinished trajectories to be resumed in subsequent iterations. This effectively eliminates generation bubbles and maximizes throughput, albeit by permitting a negligible degree of off-policy data. Fundamentally, these approaches rely on *scheduling* to mitigate long-tail inefficiency.

**Asynchronous Relaxation.** Another orthogonal line of work relaxes the synchronization constraints of RL algorithms and adopts asynchronous architectures (Fu et al., 2026; Lu et al., 2025; Han et al., 2025; Sheng et al., 2026). While hiding the long-tail latency by fully overlapping rollout with training, they deviate from the on-policy semantics. Such staleness or off-policy behavior can lead to training instability and degradation in model performance. Therefore, in this work, we focus on synchronous optimizations.

## 3. Preliminaries and Motivations

### 3.1. Notions and Preliminaries

Key notations are listed in Tab. 1. We illustrate the standard RL paradigm for LLMs using the popular GRPO (Shao et al., 2024) as an example. For each prompt $q_i$ from dataset $\mathcal{P}(Q)$, GRPO samples a group of $M$ outputs $\mathcal{Y}_i = \{o_i^j\}_{j=1}^{M}$

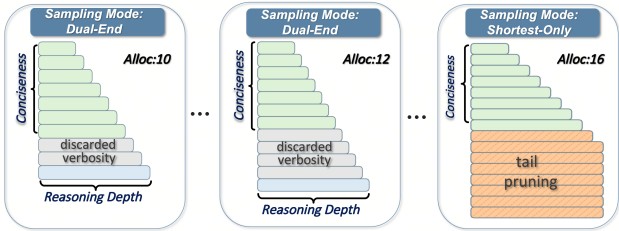

*Figure 4.* Illustration of DARTS method: adaptive sampling strategies and redundancy allocations for different prompts ($M = 8$).

reward correlation. Specifically, for a given prompt $q_i$, we compare the expected response lengths of correct ($\mathbb{E}[l_i^j | r_i^j > 0]$) and incorrect ($\mathbb{E}[l_i^j | r_i^j < 0]$) responses, identifying two distinct rollout distribution patterns.

**Pattern I: Verbose and Ineffective Tails.** In the first pattern (left of Fig. 3), the expected length of correct responses is shorter than that of incorrect ones, i.e., $\mathbb{E}[l_i^j | r_i^j > 0] \leq \mathbb{E}[l_i^j | r_i^j < 0]$. This implies that the model is capable of answering these prompts correctly with concise reasoning. However, it often exhibits unnecessary verbosity, producing chains that are correct but verbose, or trapped in incorrect logical loops. These tails incur significant computational costs while offering minimal marginal utility for learning. For such patterns, the ideal optimization objective is to encourage the model to be concise and decisive, eliminating superfluous generation, thereby enhancing efficiency.

**Pattern II: Necessary and Effective Tails.** Conversely, for complex prompts requiring extensive deep reasoning (e.g., complex mathematical proofs), we observe that correct responses are generally longer than incorrect ones, as shown in the right of Fig. 3, i.e., $\mathbb{E}[l_i^j | r_i^j > 0] > \mathbb{E}[l_i^j | r_i^j < 0]$. In this pattern, the tail is populated by high-quality, correct trajectories, whereas shorter responses often result from superficial errors or premature conclusions. Therefore, while generally encouraging conciseness, we should preserve these effective tails to ensure necessary reasoning depth.

**Motivation.** These observations inspire a novel perspective on mitigating the long-tail bottleneck: focusing on the distribution itself and shaping it towards an ideal state. We posit that an ideal rollout distribution should: (1) suppress verbose and ineffective tails to enforce conciseness and enhance efficiency; while (2) preserve necessary effective tails to maintain reasoning capabilities. To this end, we propose to actively shape the rollout distribution towards conciseness and certainty in a distribution-adaptive manner, fundamentally accelerating RL training pipeline.

## 4. Method

In this section, we present DARTS, a unified framework designed to tackle the long-tail bottleneck and accelerate

RL training pipeline via **D**istribution-aware **A**ctive **R**ollout **T**rajectory **S**haping. The core principle, *active distribution shaping*, proactively modulates the length distribution of rollout trajectories, aiming to shape the rollout towards *conciseness* and *certainty*. The framework comprises three key components: **(1) Distribution-Aware Trajectory Sampling** (Section 4.1): We explicitly expand the exploration space for each prompt via *intra-prompt redundant rollout*. From this redundantly generated candidate pool, we employ *dual-end length sampling* to selectively sample trajectories based on length distribution statistics, thereby shaping the distribution to enhance rollout efficiency. **(2) Adaptive Redundancy Allocation** (Section 4.2): To maximize the effectiveness of distribution shaping under system budget constraints, we propose a *variance-based adaptive redundancy allocation* scheme, which allocates redundancy levels to each prompt in a distribution-adaptive manner. **(3) System-Level Optimization** (Section 4.3): To further optimize framework efficiency, we propose *variance-guided tail pruning* with *proactive early stopping*, which complements our sampling strategy to accelerate rollout. Furthermore, we implement *token-level streaming* to enable fine-grained overlapping of rollout and training phase. Fig. 4 illustrates the core schemes of DARTS method.

### 4.1. Distribution-Aware Trajectory Sampling

The objective of distribution-aware trajectory sampling is to actively shape the model's rollout length distribution via selective sampling from an expanded exploration space.

#### 4.1.1. INTRA-PROMPT REDUNDANT ROLLOUT

The prerequisite for distribution shaping is the expansion of the exploration space. For each query $q_i$, we construct a *redundant candidate pool* $\mathcal{T}_i = \{o_i^1, o_i^2, \ldots, o_i^{M_i'}\}$ by explicitly over-sampling to generate $M_i'$ independent trajectories, where the candidate count exceeds the target group size ($M_i' \geq M$). This superset $\mathcal{T}_i$ ensures sufficient exploration and diversity required for the subsequent sampling process. We term this *intra-prompt redundant rollout*. Orthogonal to prompt-level over-sampling employed in tail scheduling approaches, we focus on expanding the *intra-prompt* exploration space for fine-grained trajectory sampling.

#### 4.1.2. DUAL-END LENGTH SAMPLING

From the candidate pool $\mathcal{T}_i$ of size $M_i'$, we aim to select a subset of $M$ trajectories to form the training group $\mathcal{Y}_i$ for prompt $q_i$, targeting two objectives: (1) *Enforcing Conciseness*: Filtering out ineffective verbosity to steer the model towards concise responses. (2) *Preserving Reasoning Depth*: Retaining a subset of long trajectories with necessary complex reasoning, ensuring the model maintains problem-solving capabilities. To resolve this, we propose

*dual-end length sampling*. Specifically, we construct group $\mathcal{Y}_i$ by selecting trajectories from the two extremes of the rollout length distribution within $\mathcal{T}_i$: $\mathcal{Y}_i = \mathcal{Y}_{\text{short}} \cup \mathcal{Y}_{\text{long}}$, where $\mathcal{Y}_{\text{short}}$ consists of the top-$K$ shortest trajectories to maximize conciseness, and $\mathcal{Y}_{\text{long}}$ comprises the top-$L$ longest to preserve the exploration of reasoning depth (with $K + L = M, K \gg L$). This composition ensures that the training group is dominated by concise samples while maintaining a minority of deep reasoning paths. Notably, the top-$L$ longest selection excludes incomplete and invalid trajectories reaching the system length limit.

### 4.1.3. CONNECTION TO GRADIENT DYNAMICS

This dual-end sampling strategy is grounded in a straightforward *length-distribution-aware heuristic*. We now analyze its connection to gradient dynamics. As shown in Eq. 1, the advantage $A(o_i)$ serves as a signed weighting factor for the gradient, meaning trajectories with larger $|A(o_i)|$ exert a stronger influence. Our dual-end sampling strategically constructs the training group $\mathcal{Y}_i$ to shift the average group reward $\mu(\mathcal{Y}_i)$ in Eq. 2. Since the short component dominates the group ($K \gg L$), $\mu(\mathcal{Y}_i^{\text{dual}})$ is mathematically anchored by the mean reward of the shortest trajectories ($\bar{r}_{\text{short}}$):

$$\mu(\mathcal{Y}_i^{\text{dual}}) = \frac{K\bar{r}_{\text{short}} + L\bar{r}_{\text{long}}}{K + L} \approx \bar{r}_{\text{short}} \quad (3)$$

*Proposition 1: Suppressing Verbosity.* For prompts characterized by verbosity ($\mathbb{E}[l_i^j \mid r_i^j > 0] \leq \mathbb{E}[l_i^j \mid r_i^j < 0]$), correct solutions are generally short. Consequently, $\mathcal{Y}_{\text{short}}$ captures high-reward positive samples (high $\bar{r}_{\text{short}}$). Substituting this into Eq. 3, it elevates the average group reward: $\mu(\mathcal{Y}_i^{\text{dual}}) \geq \mu(\mathcal{Y}_i^{\text{std}})$, where $\mathcal{Y}_i^{\text{std}}$ denotes the average of the standard approach. This reduces the advantage of inefficient long trajectories: $r(o_i^{\text{long}}) - \mu(\mathcal{Y}_i^{\text{dual}}) \leq r(o_i^{\text{long}}) - \mu(\mathcal{Y}_i^{\text{std}})$, which suppresses the positive advantage of verbose correct answers and amplifies the negative advantage of incorrect long tails, steering the distribution towards conciseness.

*Proposition 2: Preserving Reasoning Depth.* For complex prompts requiring depth ($\mathbb{E}[l_i^j \mid r_i^j > 0] > \mathbb{E}[l_i^j \mid r_i^j < 0]$), overly short trajectories often represent errors. Here, $\mathcal{Y}_{\text{short}}$ captures low-reward failures (low $\bar{r}_{\text{short}}$). Via Eq. 3, this results in a depressed average group reward, $\mu(\mathcal{Y}_i^{\text{dual}}) < \mu(\mathcal{Y}_i^{\text{std}})$, and valid long trajectories now gain a boosted relative advantage against this lowered baseline: $r(o_i^{\text{long}}) - \mu(\mathcal{Y}_i^{\text{dual}}) > r(o_i^{\text{long}}) - \mu(\mathcal{Y}_i^{\text{std}})$. This signal amplification ensures that necessary reasoning paths receive strong positive gradients, preserving problem-solving capabilities.

The two patterns are used only for motivational analysis; DARTS does not explicitly classify prompts into Pattern I or Pattern II during training. Instead, dual-end sampling implicitly adapts through the group reward baseline. For intermediate cases where short and long trajectories have overlapping reward distributions, the two effects coexist:

ineffective verbosity is suppressed, while useful long trajectories can still receive positive advantages.

In summary, by strategically modulating the gradient dynamics via selective sampling, our method actively steers model evolution towards a concise and compact distribution.

### 4.2. Adaptive Redundancy Allocation

We then address the upstream decision: *how should we determine the optimal redundancy budget $M_i'$ (size of $\mathcal{T}_i$) for each specific prompt $q_i$?* We observe that a uniform allocation strategy is inherently suboptimal, as it disregards the distribution characteristics across diverse prompts and fails to reconcile the trade-off between shaping effectiveness and system efficiency. To address this, we propose *adaptive redundancy allocation*, which adaptively determines the redundancy budget $M_i'$ based on the rollout distribution characteristics, ensuring computational resources are invested where they yield the highest marginal gain.

#### 4.2.1. ROLLOUT COST MODELING

We first establish an execution cost model for rollout phase. Following previous works (Zhong et al., 2025), let $\text{PTL}(d_{\text{TP}}, bsz)$ denote the per-token generation latency given a batch size $bsz$ and tensor parallel size $d_{\text{TP}}$. For a rollout batch of $M_i'$ responses $\{o_i^j\}_{j=1}^{M'}$ with sorted response lengths $l_{[1]} \leq \cdots \leq l_{[M']}$ (let $l_{[0]} = 0$), the total rollout cost $T_{\text{rollout}}$ can be formulated as the cumulative latency of piecewise generation intervals:

$$T_{\text{rollout}} = \sum_{m=1}^{M'} (l_{[m]} - l_{[m-1]}) \cdot \text{PTL}(d_{\text{TP}}, M' - m + 1) \quad (4)$$

We instantiate this model by fitting a continuous PTL curve via *profiling-based regression* (Appendix A). In data parallel, we take the maximum latency across groups as the overhead.

#### 4.2.2. VARIANCE-BASED ADAPTIVE ALLOCATION

We introduce the adaptive redundancy allocation mechanism. The redundancy budget $M_i'$ indicates the size of exploration space for sampling, presenting a trade-off between *shaping effectiveness* and *system efficiency*: insufficient redundancy ($M_i' \approx M$) constrains the exploration, reverting to standard sampling and diminishing the benefits of distribution shaping; whereas excessive redundancy ($M_i' \gg M$) incurs prohibitive rollout overhead, negating acceleration gains. Therefore, under a system budget constraint, our objective is to strategically allocate the redundancy budget into the most critical prompts with pronounced long-tail features where distribution shaping is most needed.

To navigate this trade-off, we propose a *variance-based adaptive redundancy allocation* mechanism. We identify *response length variance*, denoted as $\sigma_L^2(q_i) = $

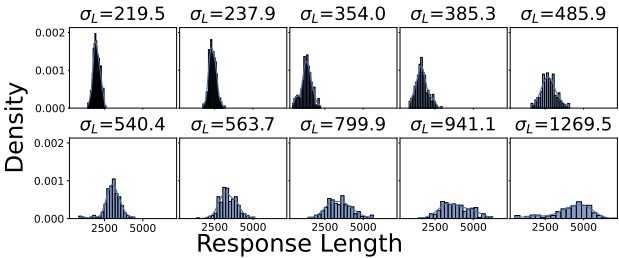

Figure 5. Observation: High variance implies more pronounced long-tail features. (Qwen2.5-Math-7B on DAPO-MATH dataset)

$\sigma^2(\{l_i^1, \ldots, l_i^{M_i'}\})$, as a robust proxy for intra-prompt long-tail severity and model uncertainty. As illustrated in Fig. 5: (1) High-variance prompts display severe long-tail distribution and elevated model uncertainty, necessitating a larger candidate pool to enable effective distribution shaping and to bridge the gap from the long-tail distribution to the target concise distribution. (2) Conversely, low-variance prompts typically feature trajectories concentrated in the short-length region, indicating a concise distribution and high model confidence. For such prompts, a minimal redundancy budget suffices. Based on these observations, we utilize the smoothed historical variance $\tilde{\sigma}_L^2(q_i)$ of prompt $q_i$, maintained via a moving average over preceding epochs, to quantify the prompt's long-tail severity and the model's uncertainty. Here we formulate the redundancy allocation as a resource optimization problem and detail our solution.

First, we introduce our *adaptive redundancy budget* scheme, which dynamically determines the optimal total system redundancy budget $M_{total}$. Specifically, we define a utility function, $U(\overline{M})$, to characterize the trade-off between algorithmic effectiveness $E(\overline{M})$ of distribution shaping and the system overhead $T(\overline{M})$ given the total budget $\overline{M}$. Let $U(\overline{M}) = E(\overline{M}) - \lambda \cdot T(\overline{M})$, where $\lambda$ is a trade-off coefficient. For system overhead, we directly utilize the rollout cost in Eq. 4 as the metric. For algorithmic effectiveness, we utilize the historical length variance $\tilde{\sigma}_L^2$ of the dataset as the metric for optimization potential. We then obtain the optimal $M_{total}$ by maximizing $U(\overline{M})$ and solving $\frac{\partial U}{\partial \overline{M}} = 0$. Detailed proof and analysis are in Appendix B.

Then, we illustrate our *variance-based adaptive allocation* scheme, which adaptively allocates redundancy level for each prompt. Given the total budget $M_{total}$, our objective is to minimize the total *distance* between the current rollout distribution and the ideal concise and compact one. We define a distance metric: $\mathcal{D}(M_i', q_i) = \text{Norm}(\tilde{\sigma}_L(q_i))/M_i'$. It is positively correlated with $\tilde{\sigma}_L(q_i)$, which denotes the long-tail severity, and inversely correlated with its redundancy allocation. $\text{Norm}(\cdot)$ represents the min-max normalization over the prompt batch to ensure robustness across different length scales. This formulation reflects the property of diminishing marginal utility: allocating additional redundancy

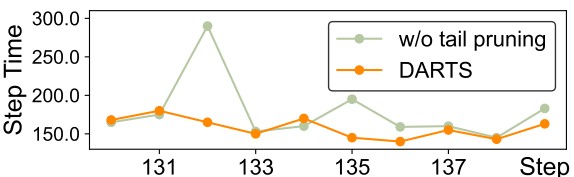

Figure 6. Efficiency comparison of w/o and w/ variance-guided tail pruning scheme. (Qwen2.5-Math-7B on 32 H20)

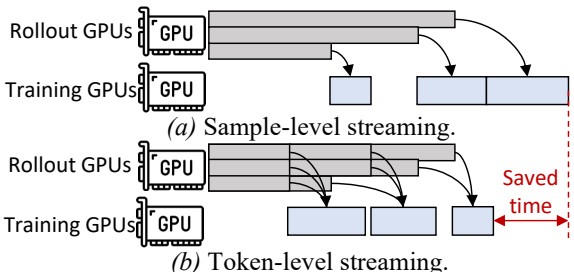

Figure 7. Timeline of sample-level and token-level streaming.

to high-variance prompts (large $\tilde{\sigma}_L(q_i)$) yields a larger reduction in distance than allocating to low-variance ones. Consequently, the optimization problem is defined as:

$$\min_{\{M_1', \ldots, M_N'\}} \quad \sum_{i=1}^{N} \frac{\text{Norm}(\tilde{\sigma}_L(q_i))}{M_i'}$$

$$\text{s.t.} \quad \sum_{i=1}^{N} M_i' = M_{total}, \quad M_i' \in \mathbb{Z}^+ \tag{5}$$

$$M_{low} \leq M_i' \leq M_{up}, \quad \forall i \in \{1, \ldots, N\}$$

where $M_{low}$ and $M_{up}$ are hyper-parameters (set to $M$ and $2M$ in our experiments) enforcing lower and upper bounds to prevent starvation or resource monopolization. Since the objective function is convex (the second derivative w.r.t. $M_i'$ is positive), this discrete optimization problem can be efficiently solved using a greedy algorithm Alg. 1. Refer to Appendix C for detailed algorithm.

## 4.3. System-Level Optimization

In this subsection, we introduce system-level optimization techniques designed to mitigate the computational bottlenecks caused by extremely long-tail distributions.

### 4.3.1. VARIANCE-GUIDED TAIL PRUNING

While generally effective, dual-end sampling incurs inefficiencies when processing highly complex prompts with extreme long tails, some of which even reach the system length limit. Strictly retaining these longest trajectories causes substantial overhead and GPU under-utilization. To address this, we propose a *variance-guided tail pruning* mechanism. We leverage the decision from our variance-

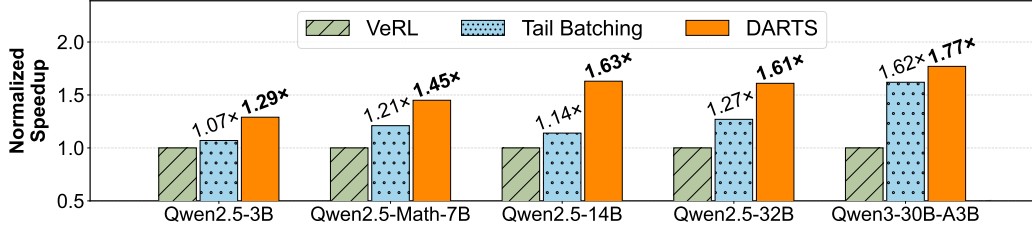

*Figure 8.* End-to-end throughput comparison for various model sizes. Speedup ratios compared to VeRL are indicated.

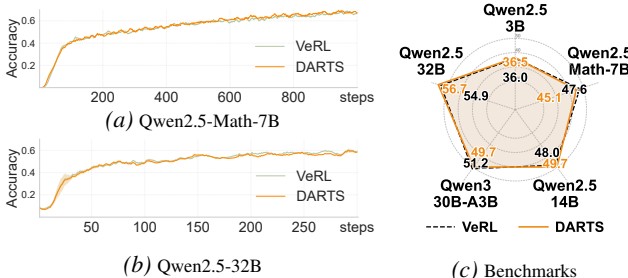

*Figure 9.* Training convergence (left) and benchmark scores (right).

based allocation (Section 4.2), and treat the saturation of the budget ($M_i' = M_{up}$) as a proxy for severe long-tail characteristics. In such cases, we dynamically switch from the dual-end sampling to *shortest-only sampling* strategy, which exclusively samples the top-$M$ shortest trajectories. This pruning is grounded in the observation that for such difficult prompts, the length distribution is already globally shifted rightward. Consequently, even the shortest trajectories typically possess sufficient reasoning depth rather than being trivial shortcuts. Experiments indicate this triggers for ∼5% of the most expensive prompts, preventing the significant throughput degradation caused by extreme outliers (Fig. 6).

**Proactive Early Stopping.** Shortest-only sampling enables a further optimization: *proactive early stopping*. In contrast to standard dual-end sampling, where the full $M_i'$ pool is required to identify the longest candidates, tail pruning simplifies to collecting only the $M$ shortest trajectories. Therefore, we implement a proactive termination signal for prompts triggering the tail pruning: the generation process is halted immediately once $M$ valid trajectories are collected, pruning the heavy-tail computation for the remaining $M_i' - M$ candidates. Overall, our complete sampling mechanism is summarized in Alg. 2 (Appendix D), integrating dual-end sampling and variance-guided pruning with early stopping.

### 4.3.2. TOKEN-LEVEL STREAMING

Following previous works (Luo et al., 2025; Zhong et al., 2025), we adopt a disaggregated architecture, decoupling rollout and training to enable flexible scaling. Existing frameworks typically employ *sample-level stream-ing* (Zhong et al., 2025), where a trajectory is transferred to the training engine immediately upon completion, thereby overlapping rollout and training. However, this coarse granularity is suboptimal in extremely long-tail scenarios. Specifically, early tokens in long trajectories must wait for the generation of the final token before training, leading to significant idle time. Furthermore, in large-scale data-parallel settings, its efficacy diminishes as the limited number of trajectories per device restricts the sample-level overlapping opportunities. To bridge this gap, we propose a finer-grained *token-level streaming* technique, as illustrated in Fig. 7 and detailed in Appendix E. Instead of awaiting full completion, we transfer data in token chunks: as soon as accumulated rollout tokens reach a predefined threshold, they are dispatched to the training engine. When the training engine accumulates sufficient token chunks, it launches the forward computation. This overlaps training the prefix with generating the suffix, maximizing efficiency in long-tail scenarios.

## 5. Experiments

### 5.1. Implementation and Experimental Setups

**Implementation.** We implement DARTS with ∼7K lines of additional Python code built upon VeRL (Sheng et al., 2025), utilizing AsyncLLMEngine of vLLM (Kwon et al., 2023) to facilitate the sampling, streaming and early stopping scheme. Specifically, we implement the sampling scheme and redundancy allocation within the VeRL main controller, and maintain the strategies dynamically within a separate Ray actor, which synchronizes states between the main controller and rollout actors to guide rollouts.

**Models and Datasets.** We conduct experiments on Qwen-series models (Yang et al., 2024; 2025) with 3B, 7B, 14B, 30B and 32B parameters, including both dense and MoE architectures. For training datasets, we use DAPO-MATH (Yu et al., 2026) for 7B-32B models and MATH-lighteval (Hendrycks et al.) for 3B model.

**Hardware Environments.** We conduct experiments on a GPU cluster with 8 nodes, with each node consisting of 8 NVIDIA H20 96GB GPUs equipped with NVLink. All nodes are interconnected by 1.6Tbps InfiniBand network. We scale the computational resources according to

model size, using 8×H20 to train Qwen2.5-3B, 32×H20 to train Qwen2.5-Math-7B, Qwen2.5-14B, and 64×H20 to train Qwen2.5-32B, Qwen3-30B-A3B.

**Baselines.** We evaluate DARTS against two representative baselines: (1) **VeRL** (Sheng et al., 2025), a state-of-the-art open-source RL framework; and (2) **Tail Batching** (from RollPacker (Gao et al., 2026)), representing the leading *prompt-level tail scheduling* approach for mitigating rollout long-tail bottlenecks. For fair comparison, we implement all approaches upon VeRL (v0.4) codebase, with vLLM (0.8.5.post1) (Kwon et al., 2023) as the rollout engine and PyTorch (2.6.0) FSDP (Zhao et al., 2023) for training, and employing disaggregated architecture (Zhong et al., 2025). All hyper-parameters and settings are kept the same across frameworks. We adopt GRPO (Shao et al., 2024) as the core RL algorithm (group size $M = 8$), incorporating optimizations from DAPO (Yu et al., 2026), e.g., clip-higher, token-level loss, and overlong reward shaping.

### 5.2. End-to-End Performance

**Throughput Comparison.** We present the end-to-end throughput comparison (averaged over 500 steps) in Fig. 8. DARTS consistently outperforms all baselines across different model scales, achieving a maximum speedup of $1.77\times$ compared to VeRL, and $1.43\times$ compared to Tail Batching. Specifically, VeRL suffers from efficiency degradation due to the long-tail bottleneck. Tail Batching mitigates this by scheduling long tails for deferred batched execution. While this alleviates idleness and achieves a $1.07\times$–$1.62\times$ speedup, such *prompt-level tail scheduling* approaches cannot fundamentally optimize the long-tail distribution itself. In contrast, DARTS actively shapes the rollout distribution towards conciseness and certainty, achieving a speedup of $1.29\times$–$1.77\times$. By guiding the model to concentrate on the short-length region, DARTS fundamentally mitigates the long-tail overhead. Notably, DARTS demonstrates superior effectiveness on larger models, which typically exhibit deeper reasoning paths and more pronounced long-tail features, offering greater optimization potential for our method.

**Training Convergence.** We conduct comprehensive convergence experiments across multiple model configurations. As shown in Fig. 9(a)(b), DARTS exhibits robust convergence stability, closely aligning with the VeRL baseline. Furthermore, we evaluate the capabilities of the trained models on five common downstream mathematical benchmarks: MATH500 (Lightman et al., 2024), GSM8K (Cobbe et al., 2021), AIME2024, AIME2025, and Olympiad (He et al., 2024). The averaged score reported in Fig. 9(c) confirms that DARTS maintains competitive accuracy without degradation, and even surpasses the baseline in specific cases. These results indicate that our distribution shaping does not compromise convergence or accuracy. This stems from

*Table 2.* Generalization evaluation on BIG-Bench Hard (BBH). DARTS improves or maintains zero-shot performance across different model scales.

| Model | VeRL | DARTS |
|---|---|---|
| Qwen2.5-3B | 34.7 | 38.7 |
| Qwen2.5-Math-7B | 56.6 | 58.8 |
| Qwen2.5-14B | 76.1 | 76.4 |
| Qwen2.5-32B | 78.1 | 84.7 |
| Qwen3-30B-A3B | 85.8 | 84.4 |

*Table 3.* Ablation study.

| Method | Speedup |
|---|---|
| VeRL | $1.00\times$ |
| + Token-Level Streaming | $1.09\times$ |
| + Distribution-Aware Sampling | $1.40\times$ |
| + Adaptive Allocation (DARTS) | $1.63\times$ |

our carefully designed sampling scheme, which effectively encourages conciseness while preserving reasoning paths.

### 5.3. Generalization Beyond Mathematical Reasoning

To further evaluate whether DARTS preserves necessary long-form reasoning beyond the main mathematical training setting, we conduct additional evaluations on BIG-Bench Hard (BBH), which contains tasks requiring logical deduction, multi-step reasoning, object tracking, and planning. As shown in Table 2, DARTS improves the zero-shot performance of most model scales compared with VeRL, while maintaining comparable performance on the remaining case. These results suggest that DARTS does not simply shorten responses at the cost of reasoning ability; instead, it removes redundant verbosity while preserving useful reasoning trajectories.

We also conduct preliminary experiments in broader domains. On multimodal reasoning with Qwen2.5-VL-3B on Geo3K, DARTS achieves a $1.20\times$ speedup. On coding generation with Phi-3-mini-3B on Eurus-2-RL-Data, DARTS achieves a $1.15\times$ speedup. The gains are smaller than those observed on larger reasoning models, likely because smaller models exhibit less pronounced long-tail rollout behavior. Nevertheless, these results indicate that the benefit of DARTS is not limited to mathematical reasoning, but applies more generally when training exhibits diverse and partially redundant intra-prompt rollout lengths.

### 5.4. Ablation Study

We conduct an ablation study to analyze the efficacy of each key component within DARTS, utilizing Qwen2.5-14B on 32 GPUs. As shown in Tab. 3, the efficiency gains primarily

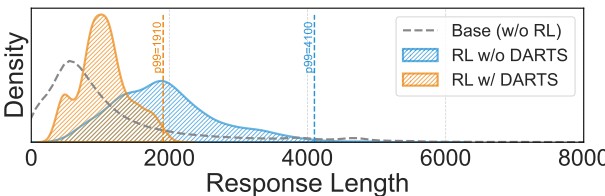

*Figure 10.* Dataset-level rollout distribution shaping effect.

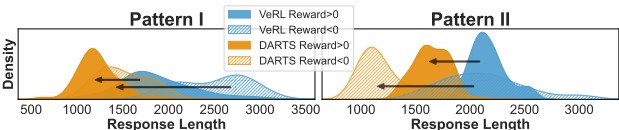

*Figure 11.* Distribution shaping effect for different prompt patterns.

stem from our distribution-aware sampling (Section 4.1) and adaptive redundancy allocation (Section 4.2). Our sampling scheme, with dual-end sampling as the core, effectively shapes the rollout distribution toward conciseness and compactness, achieving a $1.40\times$ speedup even under a naïve uniform redundancy allocation. Adaptive allocation further boosts the speedup to $1.63\times$ by adaptively allocating the exploration budget to prompts with more pronounced long-tail features, thereby maximizing both shaping effectiveness and system efficiency. Additionally, our token-level streaming (Section 4.3) also contributes a $9\%$ speedup.

### 5.5. Case Study

We present a case study using Qwen2.5-14B on DAPO-MATH dataset, comparing the rollout length distributions of models trained with the baseline (VeRL) versus DARTS. Fig. 10 presents the dataset-level average rollout distribution, where DARTS exhibits a significantly more compact distribution concentrated in the short-length region. This confirms the effective mitigation of verbose long tails, thereby reducing token consumption and improving efficiency. Furthermore, we conduct a fine-grained analysis on two prompts with distinct patterns as in Section 3.2.2. For Pattern I (Fig. 11, Left), the verbose and ineffective tails are largely eliminated, and the distributions of both correct and incorrect responses become prominently concise. For Pattern II (Fig. 11, Right), incorrect responses become concentrated and shorter, whereas the distribution of correct responses remains positioned to the right of the incorrect ones, confirming that DARTS maintains sufficient reasoning depth. Consequently, our method effectively shapes the distribution without compromising the model's capability.

### 5.6. Discussion

**Length Penalty.** Notably, DARTS employs the same length penalty settings as prior work (Yu et al., 2026). A naïve

*Table 4.* Sensitivity analysis of the dual-end sampling ratio. The reported numbers are speedups over VeRL.

| Model | 1:7 | 2:6 | 4:4 |
|---|---|---|---|
| Qwen2.5-Math-7B | $1.45\times$ | $1.43\times$ | $1.25\times$ |
| Qwen2.5-3B | $1.29\times$ | $1.35\times$ | $1.15\times$ |

*Note:* Columns denote the ratio $L$:$K$.

approach to mitigate long tails is to simply apply a larger length penalty. However, our experiments reveal that this aggressive approach causes a 2%-7% performance drop across downstream tasks, demonstrating that such forced brevity compromises reasoning capabilities. In contrast, DARTS does not modify rewards or directly penalize long responses. It reshapes the sampled GRPO group and changes relative advantages through the group reward baseline. Therefore, long responses can still receive strong positive gradients when they are correct and necessary, unlike explicit length penalties that uniformly reduce rewards for long outputs.

**Hyperparameter Sensitivity.** We further study the sensitivity of DARTS to the dual-end sampling ratio. Recall that DARTS selects the top-$K$ shortest trajectories and the top-$L$ longest trajectories, where $K + L = M$. Table 4 reports the speedup of different $L : K$ settings. DARTS remains effective under different ratios, especially when short trajectories dominate the training group. More balanced settings such as $L : K = 4 : 4$ reduce the strength of distribution shaping and therefore lead to smaller speedups, but still provide acceleration over the baseline. Therefore, we set $L = 1$ for all main experiments.

We set $M_{\text{up}} = 2M$ as a soft upper bound, which provides sufficient extra samples for distribution shaping while avoiding excessive rollout overhead. Increasing this cap brings diminishing returns because the marginal benefit of additional redundant trajectories decreases. The coefficient $\lambda$ controls the strength of adaptive allocation: a larger $\lambda$ makes the allocation more conservative, while a smaller $\lambda$ makes it closer to aggressive expansion. In practice, DARTS remains stable as long as prompts with more severe long-tail behavior receive more redundancy than low-variance prompts.

## 6. Conclusion

In this work, we aimed to tackle the long-tail bottleneck in RL rollout. We characterized the long-tail distribution in detail and introduced a novel perspective of *active distribution shaping*. To this end, we presented a framework that integrated distribution-aware trajectory sampling and an adaptive redundancy allocation scheme. Extensive experiments demonstrated its superior efficiency over SOTA frameworks.

## Acknowledgment

This work is supported by National Natural Science Foundation of China (U23B2048, 62402011, 624B1019), Fundamental and Interdisciplinary Disciplines Breakthrough Plan of the Ministry of Education of China (JYB2025XDXM108), and High-performance Computing Platform of Peking University.

## Impact Statement

This paper presents work whose goal is to advance the field of Machine Learning. Specifically, this paper aims to accelerate the LLM reinforcement learning (RL) training pipeline to advance the development of Large Language Models. By effectively mitigating the long-tail bottleneck, our method significantly reduces the computational overhead and energy consumption associated with RL training. There are many potential societal consequences of our work, none which we feel must be specifically highlighted here.

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

## A. PTL Curve Fitting

As illustrated in Section 4.2.1, we employ a *profiling-based regression* approach to construct the PTL curve. Here, we present the details. We conduct preliminary profiling on our cluster utilizing the exact same vLLM inference engine employed in the main experiments setup, across various model sizes. Specifically, under different tensor parallelism size $d_{\text{TP}}$ settings, we profile data points and measure the relationship between PTL and batch sizes $bsz$ to facilitate subsequent cost-model modeling. Empirically, we found that a three-stage piecewise linear function provides an accurate approximation of the PTL curve. Fig. 12 and Fig. 13 illustrate the fitting results for the 7B model on the H20 cluster with TP=2 and TP=4.

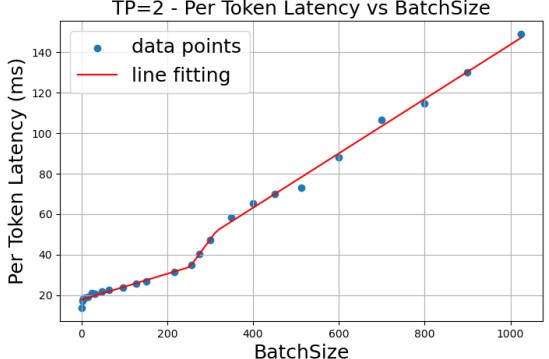
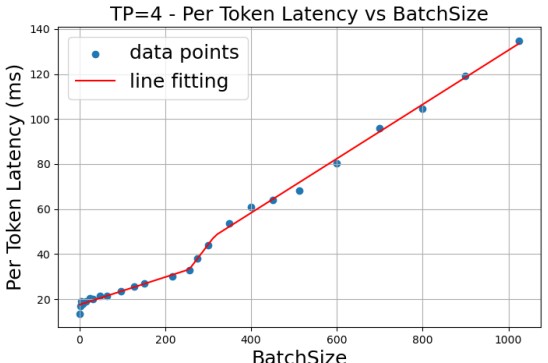

*Figure 12.* PTL curve fitting for TP=2.      *Figure 13.* PTL curve fitting for TP=4.

## B. Adaptive Redundancy Budget

As introduced in Section 4.2.2, we design an *adaptive redundancy budget* scheme to dynamically determine the optimal global redundancy budget $M_{\text{total}}$. This requires balancing algorithmic effectiveness with system latency. Therefore, we formulate a utility function as $U(\overline{M}) = E(\overline{M}) - \lambda \cdot T(\overline{M})$, where $E(\overline{M})$ represents algorithmic effectiveness and $T(\overline{M})$ denotes system efficiency. For simplicity, we employ a linear formulation to balance these two factors, with $\lambda$ serving as the trade-off coefficient. To obtain the optimal $M_{total}$, we need to find the maximum value of this function.

We now detail the definitions of $T(\overline{M})$ and $E(\overline{M})$. For the system overhead $T(\overline{M})$, we directly utilize the rollout cost model (Eq. 4) from Section 4.2.1, with the rollout latency scaling linearly with the redundancy budget:

$$T(\overline{M}) = k \cdot \overline{M} + c \tag{6}$$

where $k$ is derived from the measured PTL and cluster parameters, and $c$ denotes constant system overhead.

Next, we define the algorithmic effectiveness $E(\overline{M})$. We first introduce a metric $\rho = \frac{\tilde{\sigma}_L}{\tilde{L}}$. Here, $\tilde{\sigma}_L$ and $\tilde{L}$ denote the historical standard deviation and average of the model's response lengths, respectively. These are dataset-level statistics rather than prompt-specific values, and they evolve as training proceeds. In practice, we track them via a moving average over preceding steps. As discussed in Section 4.2, $\tilde{\sigma}_L$ serves as a robust proxy for both model uncertainty and the severity of the long-tail phenomenon. Consequently, $\rho$ quantifies the potential optimization gain from distribution shaping, as it is positively correlated with response variance. Formally, we define $E(\overline{M})$ as:

$$E(\overline{M}) = \rho_t \cdot \ln \overline{M} \tag{7}$$

where $\ln \overline{M}$ captures the diminishing marginal returns of increasing redundancy. The term $\rho_t$ (the value of $\rho$ at step $t$) acts as an amplification factor: more severe long-tail features and higher uncertainty imply a steeper effectiveness curve, justifying a larger exploration budget.

**Analytical Solution.** Our objective is to maximize the utility $U(\overline{M})$. By solving $\frac{\partial U}{\partial \overline{M}} = 0$, we can get the optimal $\overline{M}^*$:

$$\frac{\partial U}{\partial \overline{M}} = \frac{\rho_t}{M} - \lambda k = 0 \quad \Rightarrow \quad \overline{M}^* = \frac{\rho_t}{\lambda k} \tag{8}$$

This closed-form solution intuitively reveals that the optimal redundancy is proportional to uncertainty ($\rho_t$) and constrained by system cost ($k$). Finally, we constrain $\overline{M}^*$ within algorithmic needs to obtain the dynamic budget:

$$M_{\text{total}} = \text{clip}\left(\lfloor \overline{M}^* \rfloor,\ \overline{M}_{low},\ \overline{M}_{up}\right) \tag{9}$$

where $\overline{M}_{low}$ and $\overline{M}_{up}$ are set as $MN$ and $2MN$, respectively, as the lower and upper constraint of total budget. Here $N$ denotes the batch size and $M$ denotes the training group size.

**Discussion of Redundancy Budget.** $M_{total}$ represents the total redundancy budget, which defines the size of the exploration space for sampling. An insufficient budget constrains exploration and limits the shaping effectiveness of DARTS, whereas an excessively large $M_{total}$ incurs substantial system overhead. Empirically, we observe that the solved optimal $M_{total}$ is approximately $1.5NM$ on average, striking a favorable balance between algorithmic effectiveness and system efficiency.

## C. Adaptive Redundancy Allocation Algorithm

As discussed in Section 4.2.2, we propose a *variance-based adaptive redundancy allocation* scheme, which adaptively allocates redundancy budgets for each prompt based on the rollout distribution characteristics. Specifically, this scheme allocates more budget to the prompts with more pronounced long-tail features and higher model uncertainty. This is indicated via length variance, for which we maintain a historical variance estimate $\hat{\sigma}_i^2$ for each prompt $p_i$. This estimate is updated via an Exponential Moving Average (EMA) to smooth out transient noise and provide a robust signal for allocation. As illustrated in Section 4.2.2, we define the distance metric $\mathcal{D}(M_i', q_i) = \text{Norm}(\tilde{\sigma}_L(q_i))/M_i'$, and define the optimization problem as:

$$\min_{\{M_1', \ldots, M_N'\}} \quad \sum_{i=1}^{N} \frac{\text{Norm}(\tilde{\sigma}_L(q_i))}{M_i'}$$
$$\text{s.t.} \quad \sum_{i=1}^{N} M_i' = M_{total}, \quad M_i' \in \mathbb{Z}^+ \tag{10}$$
$$M_{low} \leq M_i' \leq M_{up}, \quad \forall i \in \{1, \ldots, N\}$$

Then, we solve this convex optimization problem with a greedy algorithm (Alg. 1). The allocation process operates under a global budget $M_{total}$, which is dynamically determined by the *adaptive redundancy budget* scheme (Appendix B). The complete procedure is outlined in Alg. 1. The algorithm begins by initializing the redundancy budget for all prompts to a baseline $M_{low} = M$ and computing a priority weight $w_i$ derived from the normalized historical length variance (Lines 1–2). Subsequently, it employs a greedy iterative strategy to allocate the remaining budget $M_{remain}$. In each iteration, the algorithm calculates the *marginal gain* $\Delta_i$ for all valid candidates that have not reached the upper bound $M_{up}$. As formulated in Line 4, $\Delta_i$ quantifies the expected reduction in the objective function obtained by adding additional allocation, scaled by the prompt-specific weight $w_i$. The system then identifies the prompt index $k$ that offers the maximal marginal gain (Line 5) and increments its budget $M_k'$ accordingly (Line 6). This process repeats until the total budget is exhausted. Finally, Alg. 1 outputs the optimized redundancy allocation plan $\mathcal{M}'$ (Lines 8–9).

## D. Distribution-Aware Trajectory Sampling Algorithm

As illustrated in Section 4.1 and Section 4.3.1, our *distribution-aware trajectory sampling* algorithm consists of *dual-end length sampling*, as well as *variance-guided tail pruning* with *proactive early stopping*. The complete algorithm is detailed in Alg. 2. For each prompt $q_i$ in the batch $\mathcal{B}$, the execution flow diverges based on the allocated redundancy budget $M_i'$ relative to the upper bound constraint $M_{up}$, reflecting the variance characteristics as illustrated in Section 4.2.2.

Specifically, when the allocated budget satisfies $M_i' < M_{up}$, we employ the *dual-end length sampling* strategy. We first generate a candidate set $\mathcal{T}_i$ by performing rollouts with size $M_i'$ (Line 5). To construct the final training group $\mathcal{Y}_i$ of target size $M$, we explicitly sample a subset of trajectories to shape the distribution: we retrieve the top-$K$ shortest trajectories $\mathcal{Y}_{\text{short}}$ to encourage conciseness, and the top-$L$ longest trajectories $\mathcal{Y}_{\text{long}}$ (where $L = M - K$) to maintain diversity (Lines 6–9). Notably, the top-$L$ longest selection excludes incomplete or invalid trajectories that reach the system length limit. The union of these two subsets forms the training group for the current prompt.

Conversely, when the allocated budget hits the saturation threshold ($M_i' \geq M_{up}$), this prompt is considered extremely long-tailed, and the algorithm switches to *variance-guided tail pruning* for better system efficiency. In this scenario, we

---

**Algorithm 1** Adaptive Redundancy Allocation

---

**Input:** Prompt Batch $\mathcal{B}$, Total Redundancy Budget $M_{total}$
**Output:** Redundancy Allocation Plan $\mathcal{M}'$

1: $w_i \leftarrow \text{Norm}(\tilde{\sigma}_L(q_i))$, $M_i' \leftarrow M_{low}$ for all $q_i \in \mathcal{B}$;
2: $M_{remain} \leftarrow M_{total} - N \cdot M_{low}$;
3: **while** $M_{remain} > 0$ **do**
4:     Compute marginal gains for valid candidates ($M_i' < M_{up}$):
    $\Delta_i \leftarrow w_i \cdot \left( \frac{1}{M_i'} - \frac{1}{M_i'+1} \right)$;
5:     Select optimal index: $k \leftarrow \arg\max_i \Delta_i$;
6:     Update: $M_k' \leftarrow M_k' + 1$,
    $M_{remain} \leftarrow M_{remain} - 1$;
7: **end while**
8: $\mathcal{M}' \leftarrow \{M_1', \ldots, M_N'\}$;
9: **return** $\mathcal{M}'$;

---

**Algorithm 2** Distribution-Aware Trajectory Sampling

---

**Input:** Prompt Batch $\mathcal{B}$, Redundancy Allocation Plan $\mathcal{M}'$, Target Size $M$, Dual-End Short Size $K$
**Output:** Training Group $\mathcal{Y}$ for Batch $\mathcal{B}$

1: **Initialize** $\mathcal{Y} \leftarrow \emptyset$;
2: **for** *parallel* prompt $q_i \in \mathcal{B}$ with $M_i' \in \mathcal{M}'$ **do**
3:     **if** $M_i' < M_{up}$ **then**
4:       *// Dual-End Length Sampling*
5:       $\mathcal{T}_i \leftarrow \text{Rollout}(q_i, \text{size} = M_i')$;
6:       $L \leftarrow M - K$;
7:       $\mathcal{Y}_{\text{short}} \leftarrow \text{SelectTopShortest}(\mathcal{T}_i, K)$;
8:       $\mathcal{Y}_{\text{long}} \leftarrow \text{SelectTopLongest}(\mathcal{T}_i, L)$;
9:       $\mathcal{Y}_i \leftarrow \mathcal{Y}_{\text{short}} \cup \mathcal{Y}_{\text{long}}$;
10:     **else**
11:       *// Sampling Tail Pruning with Early Stopping*
12:       $\mathcal{Y}_i \leftarrow \text{Rollout}(q_i, \text{size} = M_i', \text{early\_stop} = M)$;
13:     **end if**
14:     $\mathcal{Y} \leftarrow \mathcal{Y} \cup \mathcal{Y}_i$;
15: **end for**
16: **return** $\mathcal{Y}$;

---

perform rollouts with a *proactive early stopping* mechanism (Line 12). Instead of generating the full redundancy $M_i'$, the generation process terminates immediately once $M$ valid complete responses are collected. This effectively prunes the long-tail execution overhead for prompts that already exhibit high variance or have reached the maximum resource cap. Finally, the processed group $\mathcal{Y}_i$ is aggregated into the global training batch $\mathcal{Y}$.

## E. Details of Token-Level Streaming

As illustrated in Section 4.3.2, we propose a novel *token-level streaming* scheme to further overlap rollout generation with training.

Our implementation centers on two key architectural designs. First, within the training engine, we implement *incremental forward computation*. By maintaining a KV cache for each trajectory and employing a masking strategy that pads processed tokens, we ensure that the model performs forward passes exclusively on newly generated incremental tokens. This effectively avoids redundant computations on historical data.

Second, we adopt a *hybrid granularity management* strategy. Forward computations operate via token-level streaming: they are triggered immediately once sufficient token chunks are accumulated to maximize GPU utilization. In contrast, the reward calculation and backward pass strictly necessitate complete trajectories to maintain sample-level integrity. This design effectively overlaps the heavy forward computation latency with ongoing generation, optimizing throughput while preserving the correctness of the reward and update steps.

## F. Analysis on Response Length

Our method actively shapes the response distribution, and the temporal evolution of its effects is intuitively illustrated in Fig. 14 and Fig. 15. As shown by the orange curves (DARTS), both the average and maximum response lengths exhibit a significant reduction during the early and middle stages of training compared to the baseline (VeRL). This trend confirms that our method effectively mitigates the verbose long-tail issue early on, greatly enhancing training efficiency. Crucially, the data reveal a strategic growth in response length during the later stages (e.g., after 300 steps). This behavior demonstrates the effectiveness of our *dual-end sampling* strategy: rather than permanently constraining the model to short and concise responses, our adaptive distribution dynamically incorporates longer, complex trajectories as training progresses, enabling continuous improvement in reasoning depth.

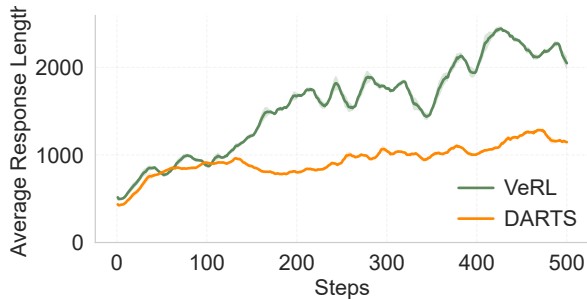

*Figure 14.* Average length comparison of Qwen2.5-14B training process on DAPO-MATH dataset.

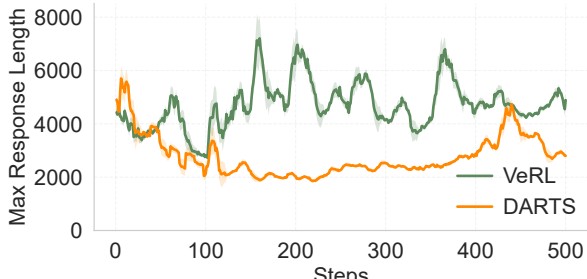

*Figure 15.* Max length comparison of Qwen2.5-14B training process on DAPO-MATH dataset.

*Table 5.* Scalability study.

| DARTS | 16 GPUs | 32 GPUs | 64 GPUs |
|---|---|---|---|
| Speedup | 1.21× | 1.44× | 1.55× |

## G. Scalability Study

To evaluate the scalability of DARTS, we conduct a scalability study and compare it against VeRL on Qwen2.5-Math-7B across varying cluster scales, ranging from 16 to 64 GPUs. As shown in Tab. 5, DARTS consistently achieves substantial acceleration across all configurations. Notably, the performance gains even amplify as the cluster size increases. This is because abundant computational resources allow for a larger redundancy exploration budget, which in turn boosts the effectiveness of our active distribution shaping.

## H. Comparison between Prompt Scheduling and Active Distribution Shaping

**The Limitation of Tail Batching.** As a representative scheduling-based mitigation strategy, Tail Batching employs a prompt-level tail scheduling scheme to enhance rollout efficiency. Specifically, it over-samples prompts and schedule long tails for deferred batched execution to mitigate the idleness and under-utilization caused by long tails. For a set of over-sampled $N'$ prompts, let $T_{group,(i)}$ denote the $i$-th order statistic of the group completion times for prompt $q_i$, $\{T_{group,1}, \ldots, T_{group,N'}\}$ (where $N' > N$). By defining the batch completion time as $T_{batch} = T_{group,(N)}$, the system successfully avoids the overhead caused by the extreme tails $(N+1, \ldots, N')$, which are deferred to a dedicated batch for later re-computation. However, we argue that Tail Batching is not a fundamental solution for the following reasons:

- **Distribution Bias:** By consistently selecting the $N$ fastest-completing groups, Tail Batching implicitly clusters prompts with inherently short generation latencies into the same training batch. This creates a bias among the training data: early batches are dominated by short, simple sequences, while later batches consist exclusively of long, complex ones. Such temporal shifts in the data distribution can lead to unstable convergence and biased gradient signals during training.
- **Persistence of Long-Tail Distribution:** Tail Batching is primarily a prompt-level scheduling heuristic; it does not fundamentally address the long-tail distribution itself. Specifically, as shown in Fig. 1, the latency distribution within a single prompt also exhibits a severe long-tail distribution, which cannot be resolve by prompt scheduling approaches. Even if a prompt is selected among the first $N$, its group completion time $T_{group}$ is still dictated by the slowest of its $M$ internal samples: $T_{group} = \max(T_1, \ldots, T_M)$. Consequently, despite scheduling optimization, such long-tail distribution persists and consumes heavy computational overhead.

**Effectiveness of DARTS.** Unlike reactive prompt scheduling in prior works, our method addresses the long-tail bottleneck by *actively shaping* the policy's response distribution during training. This creates a dual-reduction effect to resolve the problem at its root:

- **Conciseness (Mean Reduction):** Through dual-end sampling, our method encourages the model to generate more concise and information-dense responses. This effectively lowers the average response length, shifting the entire latency distribution toward the left. While Tail Batching leaves the inherent generation distribution process untouched, allowing

the model to continuously produce extremely lengthy and verbose outputs, DARTS mitigates this issue by fundamentally altering the generation tendency.

- **Certainty (Variance Reduction):** Simultaneously, our approach encourages a more concentrated output distribution. As discussed in Section 4.2.2, length variance serves as a robust proxy for both long-tail features and model uncertainty. Via variance-based optimization, the model's uncertainty is reduced, resulting in a more compact rollout distribution. This suppresses the emergence of extreme outliers within each prompt group, directly alleviating the intra-prompt bottleneck.

Ultimately, our approach provides a more principled and fundamental solution than current scheduling approaches by directly addressing the long-tail distribution at its root.

## I. Limitation

DARTS is most effective when rollouts exhibit substantial intra-prompt length diversity and partial redundancy. It may provide limited benefit for tasks whose output lengths are fixed or tightly constrained, such as translation, fixed-length style tuning, or tasks with strict word-count requirements. In extremely sparse-reward settings where rollouts provide little within-prompt discrimination, distribution shaping may also be less effective. Broader evaluation on larger multimodal and agentic models remains future work.

