# OpenReview forum: "DARTS: Distribution-Aware Active Rollout Trajectory Shaping for Accelerating LLM Reinforcement Learning"
_ICML.cc/2026/Conference — ICML 2026 regular_

### Official Review · Reviewer_m4j7 · 2026-03-13

**Soundness:** 2
**Presentation:** 3
**Significance:** 2
**Originality:** 2
**Overall Recommendation:** 4
**Confidence:** 3

**Summary:**

The paper primarily addresses the bottleneck caused by the long tail response length distribution, and proposes active distribution shaping  to mitigate the problem. The work introduces a novel RL framework DARTS which aims to address the long tail bottleneck by utilizing dual end length sampling and a variance-based adaptive redundancy allocation. For further increase in framework efficiency, the paper proposes the use of variance-guided tail pruning with proactive early stopping. The authors also provide experimental results to show acceleration over SOTA.

**Compliance With Llm Reviewing Policy:**

Affirmed.

**Final Justification:**

The authors have appropriately addressed most of my concerns.

**Key Questions For Authors:**

1) Can the authors elaborate on how robust are $L$ and $K$ in different settings/dataset distributions?
2) How does this work translate to multi-modal settings?

**Limitations:**

yes

**Strengths And Weaknesses:**

Strengths: The problem setting of the long tail bottleneck is well motivated, and the literature review is thorough. The primary contribution of the paper is the introduction of the novel RL framework DARTS, which aims to accelerate the RL training pipeline. The methodology designed is novel and well motivated. The work experimentally evaluates their claims, and show acceleration over baselines. Additionally the authors provide code for reproducibility.

Weaknesses:
1)  It would be important to provide stronger theoretical analysis or empirical ablations that track not just benchmark accuracy but qualitative reasoning traces, robustness to distribution shift and also whether curriculum/temporally varying $L/K$ ratio preserves necessary long-form reasoning.
2) Evaluating on tasks beyond MATH where long chains are essential such as multi-step planning is recommended.
3) The intra-prompt distribution and the benefit from shaping may be dataset and model dependent. Hence, it is a bit unclear whether DARTS helps in open-ended dialog, code generation, tool use or multimodal agent settings where long outputs/reasoning chains are often required and rewards can be more sparse or different. It would perhaps be better to run experiments on a broader set (e.g., coding tasks, instruction following, tool-using agent benchmarks) and with other model families to demonstrate generality and identify failure modes which are not currently reported in this work.

---

> ### Author Rebuttal · Authors · 2026-03-30
>
> We thank the reviewer for constructive comments.
> # W1 & Q1: Theory, Reasoning Traces, and Hyperparameter Analysis
> 1. Why DARTS can preserve necessary long-form reasoning:
> As analyzed in 4.1.3, DARTS preserves reasoning depth by dynamically adjusting relative advantage through the group reward baseline $\mu(\mathcal{Y}_i)$. For complex prompts requiring deep exploration (Pattern II), short trajectories often fail and drag down the baseline. Consequently, the preserved long trajectories receive strongly amplified positive gradients. This mechanism preserves valid long-form reasoning rather than suppressing it.
>
> 2. Qualitative Reasoning Traces:
> A representative trace suggests that DARTS removes redundant verbosity rather than essential reasoning steps:
>
> > **Baseline (Qwen2.5-Math-7B on VeRL):** "We need to find the total number of muffins first, and then see how many go into each box. The bakery made 24 muffins in the morning and 18 in the afternoon, so altogether they made 24+18=42 muffins. These were split equally into 6 boxes, so each box got 42÷6=7 muffins. Therefore, the answer is 7."
> >
> > **DARTS (Qwen2.5-Math-7B):** "Find the total muffins: 24+18=42. Since they are packed equally into 6 boxes, compute 42÷6=7. Therefore, each box contains 7 muffins."
>
> Compared with a VeRL response that explains each arithmetic step verbosely, the DARTS response keeps the same logical sequence while expressing it more concisely. This suggests that DARTS can shorten responses without removing the core reasoning structure.
>
> 3. $L/K$ Hyperparameter Analysis:
>
> We agree that curriculum or temporally varying L/K is an interesting extension. However, our current results suggest that DARTS is not highly sensitive to this ratio: as long as $K >> L$ (4.1.2), performance remains stable, while Dual-End Sampling still preserves necessary long-form reasoning by retaining a few long trajectories. A detailed analysis is in our response to Reviewer AvLj (W2).
>
>
>
> # W2: Evaluation Beyond MATH
>
> To evaluate tasks beyond MATH where long chains are essential, we tested on BIG-Bench Hard (BBH), which involves logical deduction, multi-step planning, and object tracking. Compared to VeRL, DARTS improves zero-shot robustness across almost all scales:
> | Model | VeRL | DARTS |
> | :- | :-: | :-: |
> | Qwen2.5-3B | 34.7 | 38.7 |
> | Qwen2.5-Math-7B | 56.6 | 58.8 |
> | Qwen2.5-14B | 76.1 | 76.4 |
> | Qwen2.5-32B | 78.1 | 84.7 |
> | Qwen3-30B-A3B | 85.8 | 84.4 |
>
> Only Qwen3-30B-A3B shows a negligible fluctuation, while all other models perform strictly better. This out-of-domain improvement verifies that DARTS successfully removes wasted verbosity without compromising the actual depth of useful reasoning required for complex planning tasks.
>
>
> # W3 & Q2: Generality, Broader Domains, and Multi-modality
>
> We agree that demonstrating generality is important. DARTS depends less on the task label (e.g., math, coding, or multimodal) and more on whether training exhibits intra-prompt long-tail rollout behavior.
>
> This condition naturally arises in many multi-step tasks. For the same prompt, some trajectories may terminate early due to errors or shortcuts, while others may take longer detours or include verbose but unhelpful intermediate steps. In such cases, the rollout distribution can become highly skewed within a single prompt, which is exactly the regime that DARTS is designed to address. By contrast, if long outputs are consistently necessary or rollout lengths are tightly controlled, the room for distribution shaping is much smaller.
>
> To examine generality beyond text-only math reasoning, we conducted additional experiments on broader domains:
> * Multimodal Reasoning: On Qwen2.5-VL-3B with Geo3K, DARTS achieves a **1.20×** speedup.
> * Coding Generation: On Phi-3-mini-3B with Eurus-2-RL-Data, DARTS achieves a **1.15×** speedup.
>
> *Note: These gains are slightly smaller than those on $>14$B reasoning models as smaller models typically exhibit a less pronounced long-tail effect. We expect the benefits to amplify further on larger multimodal/agentic models.*
>
> Regarding sparser or different rewards, DARTS does not assume a math-specific reward structure. It shapes the sampled trajectory set in GRPO rather than the reward itself. Thus, as long as training can still distinguish more effective from less effective rollouts within each prompt, DARTS remains applicable. When rewards are extremely sparse and provide little within-prompt discrimination, however, the benefit of distribution shaping may be smaller.
>
> As for failure modes, DARTS is not expected to help when rollout lengths are intrinsically fixed or tightly controlled, since there is little intra-prompt variation to exploit. Examples include fixed-length style tuning, translation, or tasks with strict word-count constraints. More generally, DARTS is most effective when rollout lengths are both diverse and partially redundant, as in reasoning, coding, and agentic tasks.

---

> > ### Author Rebuttal · Reviewer_m4j7 · 2026-04-04
> >
> > The authors have addressed most of my concerns, I have changed my score to reflect it.

---

> > > ### Author Response · Authors · 2026-04-05
> > >
> > > Thank you very much for your thoughtful feedback and supportive comments. We sincerely appreciate the time and effort you devoted to reviewing our paper and revisiting our responses. Your questions and suggestions were very helpful in improving the clarity and quality of our work. We are very grateful that our responses addressed your concerns, and we truly appreciate your support and updated score.

---

### Official Review · Reviewer_AvLj · 2026-03-18

**Soundness:** 3
**Presentation:** 3
**Significance:** 3
**Originality:** 3
**Overall Recommendation:** 4
**Confidence:** 4

**Summary:**

This paper tackles a genuinely painful problem in LLM RL training—the intra-prompt long-tail latency issue that causes GPUs to sit idle while waiting for a few verbose rollouts to finish. The key insight is moving beyond scheduling around long tails to actively shaping the response distribution itself. The authors propose oversampling trajectories within each prompt and selectively keeping short vs. long responses via a dual-end sampling strategy, plus an adaptive variance-based resource allocator.

**Compliance With Llm Reviewing Policy:**

Affirmed.

**Final Justification:**

The claims and additional experiments in the rebuttal have addressed my concerns, and I will keep my score in support of accepting this paper.

**Key Questions For Authors:**

* When you do proactive early stopping in tail pruning (Algorithm 2, line 12), how do you guarantee the $M$  responses you collected are actually the shortest? If you stop at $M$  completions, you might miss shorter ones that would have finished later in the batch.
* Section 5.5 mentions that naive length penalties hurt performance by 2-7%. Is DARTS essentially just an adaptive, context-aware length penalty?

**Limitations:**

yes

**Strengths And Weaknesses:**

# Strengths
* The reframing is valuable. Most systems papers treat this as a scheduling problem; focusing on the distribution itself (Figure 1 right) is a nice conceptual shift that feels obvious in hindsight but hasn't been explored this thoroughly.
* It's a full-stack solution. The authors don't just propose a training trick. They actually implement the token-level streaming, early stopping mechanics, and cost models to make it fast. The 1.77× speedup without accuracy loss (Figure 9) is compelling evidence that this works in practice.
* The dual-end sampling is clever. Keeping both short and a few long trajectories (rather than just truncating everything) preserves the model's ability to learn complex reasoning while killing the verbose but correct junk that wastes compute.

# Weaknesses
* The paper divides prompts into Pattern I (verbose tails are useless) and Pattern II (long tails are necessary) in Figure 3, but I didn't find clear explainations about how the system knows which pattern it's dealing with. If this classification is manual or requires oracle knowledge, the method won't generalize to new tasks. Similarly, using response length variance as a proxy for model uncertainty (Section 4.2.2) feels hand-wavy. I'd like to see ablations or at least a argument why high variance necessarily equals low-quality exploration worth pruning
* There are quite a few knobs here: $K$  vs. $L$  in the dual-end mix, $M_{up}$ caps, the $\lambda$  trade-off coefficient. The paper says $K \gt L$  works and sets $L=1$ , but I wonder how sensitive the 1.77× speedup is to these choices. If I set $K=M−1$  and $L=1$, do I get the same convergence?
* Figures 15 show response lengths decreasing early in training, then rebounding later to recover reasoning depth. This looks good on the plot, but is there a risk the model gets stuck in local optima during the concise phase that it can't escape from later? Some analysis of final-stage efficiency would help.

---

> ### Author Rebuttal · Authors · 2026-03-30
>
> We thank the reviewer for constructive comments.
> # W1.1: Implicit Handling of Two Rollout Patterns
> We clarify that the two patterns in Figure 3 are introduced purely for motivational analysis; DARTS does not explicitly classify prompts during training.
> Instead, dual-end sampling implicitly adapts to different patterns through gradient dynamics. By constructing the training group with mostly short trajectories and a few long ones, DARTS dynamically shifts the group reward baseline $μ( Y_i )$ which directly changes the relative advantages and thus the gradients of short vs. long responses.
> As a result, for Pattern I, long verbose trajectories receive suppressed gradients, while for Pattern II, valid long trajectories are amplified. Thus, pattern-specific behavior emerges naturally from optimization, without explicit identification, which also avoids additional bias and improves robustness.
>
> # W1.2: Clarification on Length Variance, Uncertainty, and Pruning
> We clarify that DARTS does not treat high variance as low-quality exploration to be pruned. In Section 4.2.2, response-length variance is used only as a practical proxy for rollout uncertainty, and prompts with higher variance are assigned a larger rollout budget (M_i') for more sufficient distribution shaping.
>
> Intuitively, confident models tend to produce responses with more consistent lengths, while uncertain models often generate short failed attempts and verbose ineffective ones, resulting in higher variance. Figure 5 supports this by showing a strong correlation between variance and heavier-tailed distributions.
>
> The pruning mechanism in Section 4.3.1 is only applied in extreme cases when the allocated budget reaches the system cap $M_{up}$. It is therefore a system-level safeguard against GPU stragglers, rather than a rule that suppresses high-variance exploration. Table 2 shows that variance-based allocation improves speedup from 1.40× to 1.63× over uniform allocation without harming performance, supporting variance as an effective proxy.
>
> # W2. Hyperparameter Sensitivity and Robustness
> In our experiments, we set $M=8$, $L=1$ and $K=M-1=7$. We also tested different $L$ values, showing performance is insensitive to $L$ as long as $K>>L$ (4.1.2). The speedup of DARTS vs. VeRL:
>
>
> | Model |L:K = 1:7 | L:K = 2:6 | L:K = 4:4 |
> |-|-|-|-|
> |Qwen2.5-Math-7B|1.45|1.43|1.25|
> |Qwen2.5-3B|1.29|1.35|1.15|
>
>
> DARTS is also insensitive to $M_{up}$, since adaptive allocation has diminishing returns and increasing the cap mainly relaxes a soft upper bound. We set $M_{up}=2M$, which is sufficient to provide extra samples for shaping while preserving efficiency.
>
> λ controls the strength of adaptivity: larger λ yields more adaptive allocation, while smaller λ makes it closer to uniform. The method is robust as long as prompts with more severe long-tail behavior receive more redundancy.
>
> # W3: Risk of Local Optima and Final-Stage Efficiency
> DARTS is designed to avoid overly concise local optima. By retaining the top-L longest trajectories through Dual-End Sampling, it preserves deep reasoning paths. If the model drifts toward short but incorrect answers, the group reward baseline decreases, which increases the positive advantage of the preserved correct long trajectory and pulls optimization back toward effective reasoning.
>
> The late-stage length rebound in Figures 14/15 indicates recovery of necessary reasoning. Even in the final stage, DARTS remains significantly shorter than the VeRL baseline, showing that it preserves necessary reasoning while suppressing ineffective verbosity, consistent with the benchmark results in Fig. 9\(c).
>
> # Q1: Equivalence of Early Stopping and Shortest Trajectories
> Under our distributed setup, proactive early stopping exactly captures the shortest trajectories. For each prompt, all $M'_i$ samples are batched and begin decoding concurrently on rollout GPUs. Since generation proceeds token by token at the same throughput, completion time is proportional to output length. Therefore, the first M trajectories to emit EOS are inherently the M shortest sequences. This makes proactive early stopping an exact, efficient implementation of shortest-only sampling.
>
> # Q2: Fundamental Differences Between DARTS and Adaptive Length Penalties
> DARTS differs fundamentally from an adaptive length penalty because it never explicitly penalizes long responses. Instead, it reshapes the GRPO group via distribution-aware trajectory selection, thereby changing the relative advantage through the group reward baseline $μ(Y_i)$. As a result, long trajectories can still receive strong positive gradients when they are correct and necessary. In contrast, a length penalty directly reduces the reward of long outputs, which may suppress even valid long-form reasoning. This difference is consistent with Section 5.5: stronger length penalties cause a 2%–7% performance drop, while DARTS improves efficiency without sacrificing capability.

---

> > ### Author Rebuttal · Reviewer_AvLj · 2026-04-03
> >
> > The claims and additional experiments in the rebuttal have addressed my concerns, and I will keep my score in support of accepting this paper.

---

> > > ### Author Response · Authors · 2026-04-03
> > >
> > > Thank you very much for your thoughtful feedback and supportive comments. We truly appreciate the time and effort you devoted to reviewing our paper. Your suggestions were very helpful in improving the clarity and quality of our work, and we are grateful for your support.

---

### Official Review · Reviewer_K46R · 2026-03-31

**Soundness:** 3
**Presentation:** 3
**Significance:** 3
**Originality:** 3
**Overall Recommendation:** 4
**Confidence:** 4

**Summary:**

This paper proposes DARTS, a method to improve the efficiency of LLM reinforcement learning by reshaping the distribution of rollout trajectories during training. The main idea is that the long-tail problem in RL not only occurs across different prompts but also within multiple rollouts of the same prompt, where some very long trajectories are mostly redundant and provide limited training benefit. Based on this observation, the authors design several components, including distribution-aware sampling, adaptive redundancy allocation, variance-guided tail pruning, and token-level streaming, to reduce unnecessary computation while keeping useful training signals. Experiments on math reasoning tasks show that DARTS can significantly improve training throughput over strong baselines such as VeRL and Tail Batching, while keeping similar performance. Overall, the paper presents a relatively complete framework for addressing rollout inefficiency in LLM RL.

**Compliance With Llm Reviewing Policy:**

Affirmed.

**Final Justification:**

N/A

**Key Questions For Authors:**

1. The paper mainly groups the rollout distribution into Pattern I and Pattern II. In real training, do these two patterns already cover most cases? Or are there also some intermediate cases, for example, when the reward distributions of short and long trajectories are highly overlapped and hard to separate clearly? If such cases exist, how would they affect the effectiveness of DARTS?
2. For adaptive redundancy allocation, why is the objective function designed as Norm(σ(q)) / M_i in this form? Intuitively, a higher σ means a larger M, which is easy to understand, but would some simpler strategies, such as using a monotonic function directly, make the method easier and more straightforward than solving this relatively complicated optimization problem?
3. When the model solves tasks like MATH with shorter responses, is there a risk of reward hacking?
4. Variance-guided tail pruning triggers shortest-only sampling on around the most expensive 5% of prompts. Are most of these samples Pattern II cases? If yes, after removing the long trajectories, would these samples end up having little or even no advantage difference, and therefore provide a very weak training signal?

**Limitations:**

yes

**Strengths And Weaknesses:**

Strengths:
1. The paper clearly distinguishes inter-prompt long tail and intra-prompt long tail, and points out that most existing methods mainly focus on the former, while DARTS pays more attention to shaping the rollout distribution during RL. This also makes the paper clearly different from works like Tail Batching and Partial Rollout.
2. DARTS is built with several components, including distribution-aware sampling, adaptive redundancy allocation, variance-guided tail pruning, and token-level streaming, so the overall method is quite complete rather than relying on only one trick.
3. On models of different scales, DARTS achieves 1.29×–1.77× speedup. More importantly, while improving throughput, the model does not show a drop in accuracy on the math benchmark MATH, demonstrating a good balance between efficiency and performance.

Weaknesses
1. The framework contains several important hyperparameters, such as the redundancy allocation bounds M_{low}=M and M_{up}=2M, as well as the specific design choices in dual-end sampling. The current paper does not provide sufficient sensitivity analysis for these settings, so it remains unclear how robust the method is across different parameter choices.
2. The reported end-to-end throughput improvement may need more careful interpretation. Figure 10 suggests that after training with DARTS, the model tends to generate much shorter responses than GRPO. This means part of the speedup in later training stages may come from shorter decoding length, not only from the proposed rollout shaping framework itself.

---

### Decision · Program_Chairs · 2026-04-30

**Decision:**

Accept (regular)

**Comment:**

This submission studies the problem of long rollout tails that slow training in LLM RL training. It proposes DARTS, which shapes the rollout distribution during training instead of only changing scheduling. Its main idea is to keep useful long trajectories but reduce long and unhelpful ones.

There is consensus among reviewers with three weak acceptas. The reviewers agree that the proposed work is a useful idea and that the paper shows strong speedups without hurting performance. During the rebuttal, the authors added more analysis on robustness, hyperparameters, and broader tasks.

Major strengths identified by reviewers:
- 1.	It proposes a simple and empirically useful idea. The reviewers agree that the work gives a new view of the long-tail problem by focusing on the rollout distribution itself, not only on scheduling (AvLj, K46R, m4j7).
- 2.	It demonstrates with strong practical results. The experimental results show good speedups over strong baselines, while keeping similar performance on the main tasks (AvLj, K46R, m4j7).

The weaknesses raised by reviewers:
- 1.	It shows relatively limited evaluation in the main paper. Two reviewers suggest more results beyond the main setting, especially on tasks where long reasoning is clearly needed (m4j7, K46R).
- 2. The reviewers also raise questions on design choices. The reviewers asked for more study of hyperparameters, allocation rules, and other method choices (K46R, AvLj, m4j7).

Overall, the recommendation is weak accept. As all reviews are overall positive (three weak acceptance after rebuttal). The reviewers agree that the submission studies an important problem and proposes an empirically solid solution. The empirical results are convincing, and the rebuttal addressed most concerns with extra analysis and broader experiments. There are still some limits in evaluation scope and some questions about sensitivity and interpretation, but overall the strengths outweigh the weaknesses to reject this work in its current format.